# Phylogenetic and Morphological Evidence Reveal Five New Species of Boletes from Southern China

**DOI:** 10.3390/jof9080814

**Published:** 2023-07-31

**Authors:** Fan Zhou, Yang Gao, Hai-Yan Song, Hai-Jing Hu, Wen-Juan Yang, Wei Zhang, Li-Yu Liao, Yi Fang, Lin Cheng, Dian-Ming Hu

**Affiliations:** 1Bioengineering and Technological Research Centre for Edible and Medicinal Fungi, Jiangxi Agricultural University, Nanchang 330045, China; zhoufan8f@126.com (F.Z.); gymyxos@126.com (Y.G.); huhaijing0504@163.com (H.-J.H.);; 2Jiangxi Key Laboratory for Conservation and Utilization of Fungal Resources, Jiangxi Agricultural University, Nanchang 330045, China; 3College of Bioscience and Bioengineering, Jiangxi Agricultural University, Nanchang 330045, China; 4Key Laboratory of Crop Physiology, Ecology and Genetic Breeding, Jiangxi Agricultural University, Ministry of Education, Nanchang 330045, China; 5Jiangxi Wuyishan National Nature Reserve Administration Bureau, Wuyishan National Nature Reserve, Shangrao 334500, Chinachl5130@163.com (L.C.)

**Keywords:** *Boletaceae*, boletales, diversity, five new taxa, morphology, phylogeny, taxonomy

## Abstract

Fungi of the order Boletales are extremely important in both ecology and economy, since most of them are ectomycorrhizal fungi, which play vital roles in maintaining forest ecosystems, water and soil protection, vegetation restoration and so on. Although previous studies have shown that this order has a very high species diversity in China, there are few reports on the species diversity of boletes in Jiangxi Province, China. Based on morphological (macroscopic and microscopic morphological characteristics) and phylogenetic analyses (ITS, LSU, and *TEF1-α* sequences), in this study, the wild boletes in Jiangxi Province were investigated, and five new species are described: *Austroboletus albus*, *Xanthoconium violaceipes*, *Xanthoconium violaceofuscum*, *Xerocomus rutilans* and *Xerocomus subsplendidus*. Descriptions and hand drawings of the new species are presented.

## 1. Introduction

The order Boletales is one of the largest groups of Basidiomycota and most of them are not only ectomycorrhizal fungi but also edible fungi. They are economically and ecologically important [1]. They could form mycorrhizal relationships with more than ten families of plants, including Betulaceae, Dipterocarpaceae, Fagaceae, Pinaceae and Salicaceae, which play vital roles in maintaining forest ecosystems, water and soil protection, vegetation restoration, and contributing to the diversification of both fungi and their host plants [2].

In the past few decades, over 500 species of Boletales, including nearly 100 new species, have been reported from China [3,4,5,6,7,8,9,10,11,12,13,14,15,16,17,18,19,20,21,22,23,24,25,26,27,28]. Before the 21st century, taxonomy relied primarily on morphological and chemotaxonomic characters, which led to the problem of homonyms and synonyms [29,30,31]. With the rapid development of molecular technology, molecular methods have been continuously applied to the taxonomy, systematics, phylogeny and biogeography of macrofungi, and the classification of boletes has made unprecedented progress. Since 2016, more than 100 new species from China have been described [1,2,32,33,34,35,36,37,38,39,40,41,42,43,44,45], but few of them have been reported in Jiangxi Province [2,36,38,44,46,47].

Jiangxi has the second highest forest coverage in China, which is dominated by evergreen broad-leaved forests with a subtropical warm and humid monsoon climate and abundant rainfall. The climatic conditions and vegetation types are very suitable for the growth and reproduction of macrofungi. It is predicted that there are more than 12,000 species of fungi in Jiangxi Province, including more than 1000 species of macrofungi [48].

Until 2016, 659 species had been reported in Jiangxi Province [49], and there are many macrofungi waiting to be discovered. In this study, five new species of boletes were collected and described during the investigation of the macrofungi in Wuyishan National Nature Reserve in Jiangxi Province.

## 2. Materials and Methods

### 2.1. Sample Collection

In this study, 140 boletes specimens were collected in Jiangxi Province. All specimens were completely dug out with a shovel after taking photos of the habitat and related characteristics of the basidiomata, then a part of the context was cut from the connection between pileus and stipe and dried with silica gel to extract DNA, and the remaining specimens were dried at 41–56 °C by a food-grade fruit dryer.

### 2.2. Morphological Studies

Twelve specimens of five new boletes species were stored in the Herbarium of Fungi of Jiangxi Agricultural University (HFJAU). The macroscopic morphological characteristics mainly come from field records and photographs of basidiomata. Color codes were obtained from Kornerup & Wanscher [50]. Micromorphological descriptions were based on dried materials rehydrated in 5% KOH and stained with ammoniacal Congo red. Freehand sections were performed by using a Nikon SMZ1270 (NIKON Corporation, Japan) stereomicroscope, following the standard method described in previous studies [19,22,51,52]. Microstructures were observed with a Nikon Y–TV55 (NIKON Corporation, Japan) compound microscope. Basidiospores with special structure were examined with a ZEISS EVO18 (GER) scanning electron microscope (SEM).

The number of measured basidiospores is given as n/m/p, which means that the measurements were created on n basidiospores from m basidiomata of p collections. Dimensions of basidiospores are given as (a)b–c(d), where the range b–c represents a minimum of 90% of the measured values (5th to 95th percentile), and extreme values (a and d), whenever present (a < 5th percentile, d > 95th percentile), are in parentheses. Q represents the ratio of length/width of the spores. Qm refers to the average Q of basidiospores ± sample standard deviation [53].

### 2.3. DNA Extraction, Amplification, and Sequencing

Whole-genome DNA was extracted from approximately 0.5 g of dried specimens by optimized CTAB method [54,55]. The primer pairs ITS1/ITS4 were used to amplify the ITS region [56], LR0R/LR5 were used to amplify the large subunit ribosomal region (nrLSU) [57,58] and TEF1-983F/TEF1-1567R were used to amplify the translation elongation factor 1-α region (*TEF1-α*) [24,59].

The PCR reaction was under the following conditions: 94 °C for 4 min, then 35 cycles of 94 °C for 60 s (50 °C for ITS, 53 °C for nrLSU and *TEF1-α*) for 40 s, and 72 °C for 80 s, followed by a final extension step of 72 °C for 8 min [24].

PCR products were detected by NanoDrop One (Thermo Fisher Scientific, Waltham, MA, USA) and 1% agarose gels, and then sequenced by TSINGKE Biological Technology (Hunan, China) using the same primers used for PCR amplification.

### 2.4. Phylogenetic Analyses

Sequences obtained by sequencing were visualized and edited with BioEdit v7.0.9 [60], and then submitted to NCBI online website for Nucleotide BLAST search (https://blast.ncbi.nlm.nih.gov/Blast.cgi, (accessed on 3 October 2022)) to determine which genus the specimens belonged to. Based on BLAST results, all available nrLSU and *TEF1-α* sequences were downloaded from NCBI, and were used to identify the relationships among all of our samples and known related species in the GenBank, and to evaluate the variability of the *TEF1-α* intron region by single-gene analysis and a later polygene phylogenetic analysis [31]. The data ultimately used for phylogenetic analysis are shown in Table 1.

Sequence datasets were aligned on the online website MAFFT version 7 (http://mafft.cbrc.jp/alignment/server/, (accessed on 6 October 2022)) [61], the concatenation of the sequences of the two or three genes was completed in PhyloSuite [62]. The gene fragments of some taxa that could be found or sequenced were regarded as missing data. The intron regions of protein-coding genes were retained in the final analyses [27]. These datasets were then analyzed using RAxML version 8 [63] and MrBayes v3.2 [64] for maximum likelihood (ML) and Bayesian inference (BI), respectively. For ML analyses, under GTRGAMMAI model [65], nonparametric bootstrap analysis with 1000 repetitions [66] was used to determine the statistical support of phylogeney. For BI analyses, substitution models of partition in the datasets were determined using the Bayesian information criterion (BIC) implemented in PartitionFinder 2 [67]. Two or four MCMC runs and trees were sampled every 1000 generations. At the end of the runs, the average deviation of split frequencies was below 0.01. Other parameters were kept at their default settings. Trees were summarized, and posterior probabilities (PPs) were calculated after discarding the first 25% of generations as burn-in. Figtree v1.4.4 was used for visualization of phylogenetic analysis results. Branches that received bootstrap support for maximum likelihood (ML) and Bayesian posterior probabilities (BPP) greater than or equal to 50% (BS) and 0.95 (PP) are shown above.

## 3. Results

### 3.1. Molecular Phylogenetic Results

For *Austroboletus*, six new sequences (two of ITS, two of nrLSU, two of *TEF1-α*) from two collections were generated. For the combined dataset, HKY + I + G, TIM + I + G and TRNEF + G were evaluated as best-fit substitution models for the ITS, nrLSU and *TEF1-α* partitions, respectively. The ITS dataset consisted of 43 taxa and 1097 characters. The nrLSU dataset included 60 taxa and 926 characters. The *TEF1-α* dataset comprised 23 accessions and 623 characters. The combined nuclear dataset (ITS + nrLSU + *TEF1-α*) contains 126 sequences with 2646 nucleotide sites, and the alignment was deposited in TreeBASE (S30554). The maximum likelihood and Bayesian phylograms have no conflict in topology, the ML trees with both BS and PP values are shown in Figure 1, the green background represents the new species identified in this study, the blue represents other known species of the genus, and the red represents the outgroup.

For *Xanthoconium*, ten new sequences (five of nrLSU, five of *TEF1-α*) were generated. For the combined datasets, TRNEF + I + G and TRNEF + G were the best-fit substitution models for the nrLSU and *TEF1-α* partitions, respectively. The nrLSU dataset included 23 taxa and 883 characters. The *TEF1-α* dataset comprised 20 accessions and 633 characters, and the alignment was deposited in TreeBASE (S30556). The Maximum likelihood and Bayesian phylograms have no conflict in topology, the ML trees with both BS and PP values are shown in Figure 2, the green background represents the new species identified in this study, the blue represents other known species of the genus, and the red represents the outgroup.

For *Xerocomus*, ten new sequences (five of nrLSU, five of *TEF1-α*) were generated. For the combined dataset, TRN + I + G and TRNEF + I + G were the best-fit substitution models for the nrLSU and *TEF1-α* partitions, respectively. The nrLSU dataset included 45 taxa and 897 characters. The *TEF1-α* dataset comprised 36 accessions and 619 characters, and the alignment was deposited in TreeBASE (S30557). The Maximum likelihood and Bayesian phylograms have no conflict in topology, the ML trees with both BS and PP values are shown in Figure 3, the green background represents the new species identified in this study, the blue represents other known species of the genus, and the red represents the outgroup.

### 3.2. Taxonomy

*Austroboletus albus* F. Zhou, H.Y. Song and D.M. Hu, sp. nov. (Figure 4, Figure 5 and Figure 6).

Mycobank: MB844812.

Etymology—Latin “*albus*” refers to the color of pileus, which is white.

Diagnosis: *Austroboletus albus* is characterized by a gray-green reticulate texture pileus, emerald green when young, white to pale brown when mature and an uneven stipe, similar to a gully, covered with obvious white to yellowish white reticulation, subfusiform to amygdaliform basidiospores measuring 13–17 × 6.5–7.5 μm, surface with intricate reticulum or irregular pits, becoming shorter to nearly smooth toward both poles, yellow to golden yellow in KOH.

Holotype: China. Jiangxi Province: Shangrao City, Yanshan County, Wuyishan Nature Reserve, 117°42′56″ E, 27°48′57″ N, elev. 900 m, 23 August 2021, F. Zhou, HFJAU12002 (WYS215).

Description: Basidiomata are small to medium-sized. Pileus 2–5 cm diam, hemispherical when young, becoming convex to subhemispherical with maturity, surface dry, emerald green when young, and white (5A1) to pale brown (4A2) when mature, with gray-green (26B3) reticulate texture, margin decurved, marginal veil white (5A1), serrated to irregular. Context is 0.5–1.6 cm in thickness in the center of the pileus, white (5A1), no change when injured. Hymenophore adnate to slightly depressed around the stipe; pores circular to angular, 1–2 per mm, white (7A1) when young and pinkish white (11A2) to pink (12B3) when old; tubes up to 1.3 cm long, pinkish (9A2), no change in color when injured. Stipe 4–12 × 0.5–1.1 cm, central, cylindrical to clavate, solid; surface dry, uneven, similar to a gully, covered with obvious white (3A1) to yellowish white (1A2) reticulation. Context is white (2A1), no change when injured; basal mycelium white (2A1). Odor indistinct.

Basidia 30–40 × 10.5–12.5 μm, thin-walled, clavate, four-spored; sterigmata 2–5 μm long. Basidiospores [40/2/2] (12–)13–17(–17.5) × (6–)6.5–7.5(–8.5) μm, Q = (1.63–)1.75–2.2(–2.5), Qm = 2.10 ± 0.21, including ornamentation, subfusiform to amygdaliform and slightly angular; surface with intricate reticulum or irregular pits becomes shorter to nearly smooth toward the both poles and yellow to golden yellow in KOH. Hymenophoral trama boletoid is composed of thin- to slightly thick-walled hyphae, 3–8 μm wide and hyaline in KOH. Cheilocystidia are 36–67 × 7.4–10.3 μm, subfusiform to subfusoid-mucronate, sometimes narrowly mucronate, rostrate, thin-walled and hyaline to grayish brown in KOH. Pleurocystidia are absent. Pileipellis is a trichoderm 65–160 μm thick, composed of hyaline to grayish yellow in KOH, thin-walled, and 3–7.5 μm diam; terminal cells are 22–60 × 3–10 μm, clavate or subterete, with an obtuse apex. Pileus trama is composed of thin-walled hyphae 4–11 μm in diameter. Clamp connections are absent in all tissues.

Habitat: Solitary or group on the wet ground under mixed forests of Fagaceae (*Fagus longipetiolata*) and Pinaceae (*Tsuga chinensis* and *Pinus massoniana*).

Distribution: Jiangxi Province, China.

Additional specimens examined: China, Jiangxi Province: Shangrao City, Yanshan County, Wuyishan Nature Reserve, 117°42′54″ E, 27°48′59″ N, elev. 1050 m, 23 August 2021, F. Zhou, HFJAU12001 (WYS222).

*Xanthoconium violaceipes* F. Zhou, H.Y. Song and D.M. Hu, sp. nov. (Figure 7 and Figure 8).

Mycobank: MB847090.

Etymology—Latin “*violaceipes*” refers to the color of stipe, which is purple.

Diagnosis: *Xanthoconium violaceipes* have a hemispherical to convex to applanate pileus, purple-black to purple when young, brownish green when mature, a obvious dark purple stripes, slightly interwoven near the pileus. Basidiospores measure approximately 12.5–16.5 × 4–5.5 μm, ellipsoid to subfusiform to cylindrical, pileipellis is 60–156 μm thick and stipitipellis is 55–110 μm thick.

Holotype: China, Jiangxi Province: Shangrao City, Yanshan County, Wuyishan Nature Reserve, 117°45′44″ E, 27°50′1″ N, elev. 1880 m, 23 August 2022, F. Zhou, HFJAU12006 (WYS642).

Description: Basidiomata are medium-sized to large. Pileus 1–10 cm in diameter, initially hemispherical, becomes convex to applanate with maturity, surface is dry, purple-black (11F1) to purple (11B3) when young, brownish green (30B6–30E6) when mature, margin is lighter and deeper in the center. Context is 0.3–0.7 cm in thickness in the center of the pileus, white (1A1) and does not change when injured. Hymenophore adnate around the stipe are quite crowded; pores are circular to angular, 2–3 per mm, white (1A1) when young and yellow (1A5) to yellow-brown (1B5) when old; tubes up to 1 cm long are white (2A1) to pale brown (2B2) and do not change in color when injured. Stipe 2–10.5 × 0.6–1.5 cm, central, cylindrical, solid; surface dry, with obvious dark purple (16D3) stripes, slightly interwoven near the pileus; context is white (16A1) to light pink (16A2), no reaction when bruised; basal mycelium white. Odor indistinct.

Basidia are 20–36 × 11.5–16 μm, thin-walled, clavate and four-spored; sterigmata are 3–7 μm long, colorless to hyaline in KOH. Basidiospores [40/2/2] (11.5–)12.5–16.5(–17) × (3.5–)4–5.5(–6) μm, Q = (2.18–)2.27–3.33(–3.71), Qm = 3.00 ± 0.35, ellipsoid to subfusiform to cylindrical, slightly thick-walled (up to 0.5 μm), golden yellow in KOH, smooth. Hymenophoral trama are composed of subparallel hyphae 5–12 μm broad, colorless to hyaline in 5% KOH. Cheilocystidia are 35–55 × 8–12 μm, clavate to subfusiform, rarely mucronate, rostrate, thin-walled and colorless to hyaline in KOH. Pleurocystidia are 45–80 × 21–24 μm, clavate and cystiform. Pileipellis is a trichoderm 60–156 μm thick, composed of hyaline hyphae in KOH, thin-walled, and 4–13 μm in diameter; terminal cells are 14–46.5 × 8–18.5 μm, clavate to pyriform or cystidioid, with obtuse apex. Pileus trama is composed of thin-walled hyphae 5–12 μm in diameter. Stipitipellis is 55–110 μm thick, clavate with an obtuse apex, terminal cells (18–25 × 5–11 μm), and colorless in KOH. Stipe trama is composed of parallel hyphae 5–10 μm wide. Clamp connections are absent in all tissues.

Habitat: Scattered on soil in subtropical forests of Fagaceae, including *Lithocarpus* spp., *Castanopsis* spp. and *Quercus* spp.

Distribution: Jiangxi Province, China.

Additional specimens examined: China, Jiangxi Province: Shangrao City, Yanshan County, Wuyishan Nature Reserve, 117°45′42″ E, 27°50′3″ N, elev. 1750 m, 1 August 2021, F. Zhou, HFJAU12004 (WYS121); Shangrao City, Yanshan County, Wuyishan Nature Reserve, 117°45′44″ E, 27°50′3″ N, elev. 1840 m, 23 August 2022, F. Zhou, HFJAU12005 (WYS619).

*Xanthoconium violaceofuscum* F. Zhou, H.Y. Song and D.M. Hu, sp. nov. (Figure 9 and Figure 10).

Mycobank: MB847089.

Etymology—Latin “*violaceofuscum*” refers to the color of pileus, which is purple-brown.

Diagnosis: *Xanthoconium violaceofuscum* is characterized by purplish brown pileus, surface normally densely covered with dark reddish brown tomentose scales, and the context of pileus is pale purple, changing to blue first, then black when injured, the tubes are purplish brown to yellow to yellowish brown, change from blue to black, and has a cylindrical or clavate stipe, purple-brown to black when young and black when mature. Basidiospores are 7–9 × 3.7–5 μm, ellipsoid to subfusiform and golden yellow in KOH.

Holotype: China, Jiangxi Province: Shangrao City, Yanshan County, Wuyishan Nature Reserve, 117°45′43″ E, 27°50′29″ N, elev. 1700 m, 27 August 2022, F. Zhou, HFJAU12007 (WYS717).

Description: Basidiomata are medium-sized to large. Pileus is 3–13 cm diameter, hemispherical to subhemispherical to applanate, surface is dry, purplish brown (7C2), surface is normally densely covered with dark reddish brown (7E3) tomentose scales. Context are up to 1.5 cm in thickness in the center, pale purple (7B2), change to blue (19B5) first, then black (7E1) when injured. Hymenophore adnate around the stipe, scattered; pores circular to angular, 1–2 per mm, purplish brown (7D4) when young and yellow (5B5) to yellowish brown (5D5) when old; tubes are up to 1.5 cm long, purplish brown (7D4) to yellow (5B5) to yellowish brown (5D5), change to blue (19B5) to black (7E1) when injured. Stipe is 3.5–7 × 1–3 cm, central, cylindrical or clavate, solid; surface is dry, purple-brown (7E3) to black (7E1) when young and black (6F1) when mature. Context is pale purple-brown (6D2), changes to black (6F1) when bruised; basal mycelium is purple-gray (6D1). Odor indistinct.

Basidia are 22–34 × 7.5–13 μm, thin-walled, clavate and four-spored; sterigmata are 2–6 μm long. Basidiospores are [40/2/2] 7–9(–11.5) × (3.5–)3.7–5(–6) μm, Q = (1.67–)1.75–2.29(–2.36), Qm = 1.97 ± 0.19, ellipsoid to subfusiform, thin-walled and golden yellow in KOH, smooth. Hymenophoral trama is composed of subparallel hyphae 6–10 μm broad, yellowish white to hyaline in 5% KOH. Cheilocystidia are 37.5–64 × 7–10 μm, subfusiform to subfusoid-mucronate, sometimes narrowly mucronate, rostrate, thin-walled, hyaline to grayish brown in KOH. Pleurocystidia are 33–118 × 12–23 μm, clavate or cystiform. Pileipellis is a trichoderm 30–65 μm thick, composed of hyaline to grayish yellow in KOH, thin-walled, 7–12 μm in diameter; terminal cells are 30–65 × 10–13.5 μm, clavate or subterete, with an obtuse apex. Pileus trama is composed of thin-walled hyphae 7–12 μm in diameter. Stipitipellis ca. is 150 μm thick, with clavate and cystiform terminal cells (37–74 × 8–15.5 μm), colorless and sometimes yellowish brown in KOH. Stipe trama is composed of parallel hyphae 6–12 μm wide. Clamp connections are absent in all tissues.

Habitat: Scattered on soil in subtropical forests of Fagaceae, including *Lithocarpus* spp., *Castanopsis* spp. and *Quercus* spp.

Distribution: Currently only known from Jiangxi Province, China.

Additional specimens examined: China, Jiangxi Province: Shangrao City, Yanshan County, Wuyishan Nature Reserve, 117°45′13″ E, 27°50′14″ N, elev. 1550 m, 27 August 2022, F. Zhou, HFJAU12008 (WYS724).

*Xerocomus rutilans* F. Zhou, H.Y. Song and D.M. Hu, sp. nov. (Figure 11 and Figure 12).

Mycobank: MB847091.

Etymology—Latin “*rutilans*” refers to the color of pileus, which is ochre.

Diagnosis: The pileus of *Xerocomus rutilans* is purple to ochre when young, brownish yellow when mature, the pores are yellow when young and yellowish brown when old and the stipes are covered with white tomentose scales, ellipsoidal to elongated to fusiform basidiospores measuring 8.5–11.5 × 4–5.5 μm, pale yellow to yellow brown in KOH.

Holotype: China, Jiangxi Province: Shangrao City, Yanshan County, Wuyishan Nature Reserve, 117°46′52″ E, 27°59′31″ N, elev. 430 m, 27 August 2022, F. Zhou, HFJAU12013 (WYS693).

Description: Basidiomata are medium-sized to large. Pileus is 2.9–9 cm in diameter, initially hemispherical, becomes convex to applanate with maturity and the surface is dry, purple (11A2) to ochre (9B4) when young, brownish yellow (5B6) when mature. Context is up to 1 cm in thickness in the center of the pileus, white (5A1) and does not change when injured. Hymenophore adnate to slightly depressed around the stipe, quite crowded; pores iiragular, 2–3 per mm, yellow (2A6) when young and yellowish brown (5C6) when old, slightly higher around the stipe; tubes are up to 1 cm long, golden (2A7) to yellowish brown (5C6) and do not change in color when injured. Stipe is 4–9 × 1–3 cm, central, cylindrical, solid, dry and covered with white (2A1) tomentose scales; context is white (2A1) and has no reaction when bruised; basal mycelium is white (1A1). Odor indistinct.

Basidia are 26–45 × 9–11.5 μm, thin-walled, clavate and four-spored; sterigmata are 3.5–6 μm long. Basidiospores are [40/2/2] (8–)8.5–11.5(–12) × (3.5–)4–5.5(–6) μm, Q = (1.83–)2–2.56(–2.75), Qm = 2.27 ± 0.23, ellipsoidal to elongated to fusiform, thin-walled, pale yellow to yellow-brown in KOH, smooth. Hymenophoral trama is composed of subparallel hyphae 5–13 μm broad, colorless to hyaline in KOH. Cheilocystidia are 22–36 × 9–12 μm, common, clavate to subclavate, thin-walled and hyaline to grayish brown in KOH. Pleurocystidia are 28–48 × 8–11 μm, common, clavate to subfusiform or cystidioid. Pileipellis is a trichoderm 60–140 μm thick composed of hyaline to yellow hyphae in KOH, thin-walled, and 6–11 μm in diameter; terminal cells 26–81 × 6–10 μm, clavate to cylindrical. Pileus trama is composed of thin-walled hyphae 5–12 μm in diameter. Stipitipellis is up to 100 μm thick, with clavate to ventricose terminal cells (25–47 × 7–16 μm), hyaline to grayish brown in KOH. Stipe trama is composed of parallel hyphae 4–12.5 μm wide. Clamp connections are absent in all tissues.

Habitat: Solitary or group on the wet ground under mixed forests of Fagaceae (*Fagus longipetiolata*) and Pinaceae (*Tsuga chinensis* and *Pinus massoniana*).

Distribution: Jiangxi Province, China.

Additional specimens examined: China, Jiangxi Province: Shangrao City, Yanshan County, Wuyishan Nature Reserve, 117°46′50″ E, 27°59′25″ N, elev. 400 m, 6 July 2022, F. Zhou, HFJAU12012 (WYS531).

*Xerocomus subsplendidus* F. Zhou, H.Y. Song and D.M. Hu, sp. nov. (Figure 13 and Figure 14).

Mycobank: MB847092.

Etymology—Latin “*subsplendidus*” refers to the color of pileus, which is yellowish brown.

Diagnosis: *Xerocomus subsplendidus* has a subhemispherical to applanate to infundibulate pileus, and the surface is always covered with yellowish brown tomentose when young, cracking into brown to dark brown scales with age, always has a crooked stipe, covered with pale brown to reddish brown tomentose in the upper part, and pores and tubes change to blue when injured, has ellipsoid to subfusiform to cylindrical; basidiospores are 9–15.5 × 4–5.5 μm, pale yellow to yellow-brown in KOH.

Holotype: China, Jiangxi Province: Shangrao City, Yanshan County, Wuyishan Nature Reserve, 117°46′7″ E, 27°50′36″ N, elev. 1750 m, 27 August 2022, F. Zhou, HFJAU12010 (WYS704).

Description: Basidiomata are small to medium-sized. Pileus is 2.2–5.6 cm in diameter, subhemispherical to applanate to infundibulate, initially earth-yellow (6B5) to yellowish brown (6B7), and brown (6C7) to dark brown (6E5) later and have lighter margins; surface dry, always covered with yellowish brown (6C5) tomentose when young, cracking into brown (6A4) to dark brown (6E5) scales with age. Context is white (6A1), up to 0.8 cm in thickness in the center of the pileus, and does not change when injured. Hymenophore is slightly depressed and decurrent around the apex of the stipe, quite scattered; pores are irregular, 1 per mm or less, yellow (3A6), changes to blue (22D5) when bruised; tubes are up to 1 cm long, golden (2A6) to yellowish brown (2D7), changes to blue (22D5) when injured. Stipe is 2.5–5 × 0.4–0.7 cm, central, cylindrical or crooked, solid; covered with pale brown (2B3) to reddish brown (9B4) tomentose in the upper part; context is white (3A1) to pale brown (3A2), no reaction when bruised; basal mycelium white. Odor indistinct.

Basidia are 27–38.5 × 9–11 μm, thin-walled, clavate and four-spored; sterigmata are 3–7 μm long. Basidiospores are [60/3/3] 9–15.5(–16) × 4–5.5(–6) μm, Q = (1.82–)2–3.11(–3.56), Qm = 2.40 ± 0.32, ellipsoid to subfusiform to cylindrical, pale yellow to yellow-brown in KOH and smooth. Hymenophoral trama is composed of subparallel hyphae 5–13 μm broad, colorless to hyaline in KOH. Cheilocystidia are 39–65 × 8.5–12 μm, lanceolate, clavate to ventricose, thin-walled and hyaline to grayish brown in KOH. Pleurocystidia are 31–82 × 10–13 μm, common, similar to cheilocystidia. Pileipellis is a trichoderm 70–230 μm thick composed of hyaline hyphae in KOH, thin-walled, 5–13 μm in diameter; terminal cells are 27.5–54.5 × 10–16 μm, clavate or cystidioid. Pileus trama is composed of thin-walled hyphae 5–12 μm diam. Stipitipellis is 45–140 μm thick, with clavate to pyriform terminal cells (18–28.5 × 8.5–15 μm) and hyaline to grayish brown in KOH. Stipe trama is composed of parallel hyphae 6–18 μm wide. Clamp connections are absent in all tissues.

Habitat: Solitary or group on the wet ground under mixed forests of Fagaceae (*Fagus longipetiolata*) and Pinaceae (*Tsuga chinensis* and *Pinus massoniana*).

Distribution: Jiangxi Province, China.

Additional specimens examined: China, Jiangxi Province: Shangrao City, Yanshan County, Wuyishan Nature Reserve, 117°45′56″ E, 27°50′22″ N, elev. 1700 m, 23 August 2021, F. Zhou, HFJAU12009 (WYS232); Shangrao City, Yanshan County, Wuyishan Nature Reserve, 117°45′43″ E, 27°50′29″ N, elev. 1740 m, 27 August 2022, F. Zhou, HFJAU12011 (WYS718).

## 4. Discussion

The macro and micromorphology of *Austroboletus albus* conforms to the characteristics of *Austroboletus*. It shared similar morphological characteristics with *A. subflavidus*, *A. albidus* and *A. roseialbus*. However, *A. subflavidus* has larger basidia [27–49(51) × 12–19 μm] and basidiospores [(13.1)15.9 ± 1.15(19.5) × (5.5)7.0 ± 0.58(8.7) μm], slightly shorter and coarser stipe [(2.9)4.5–7.5(10.2) × (0.4)0.6–1.8(2.0) cm], pileus surface often cracks with matures [68]; *A. albidus* can be easily differentiated by wider basidia (28–39 × 14–20 μm), shorter stipe (4–7 × 0.4–0.6 cm), and the color of pileus is white to cream when young, cream to grayish yellow when mature, covered with light orange to brownish nubby squamules [2]; *A. roseialbus* has smaller basidiospores (11.2–14 × 6.3–7 μm), wider basidia (28–35 × 10–14 μm), shorter stipe (7–8 cm), and the color of the pileus is whitish with obvious pale pink tinges [69].

Phylogenetically, the specimens we collected formed an independent clade with strong support (BS = 100%, PP = 1), and were closed to *A. albovirescens* (HKAS:59624 and HKAS74743) [1,2,31] with high statistical support (BS = 100%, PP = 1). However, *A. albovirescens* was described by Li and Yang [2], which is characterized by the matte green to gray-green pileus, the ornamentation of basidiospores with unequal pits, and the ixocutis pileipellis. Meanwhile, they formed a large clade with *A. mutabilis* (BS = 100%, PP = 1), a species recorded in Australia [70]. However, *A. mutabilis* can be easily distinguished by shorter basidiospores (11.9–14.7 × 4.9–7 µm) and basidia (25–35 × 10–13 µm), and the color of the pileus is dark red to brownish red when young, soon fading to brownish orange, becoming orangish yellow, then eventually yellowish.

Morphologically, *Xa. violaceipes* is similar to *Boletus violaceofuscus* and *Xa. separans*. Howener, *B. violaceofuscus* has longer and narrower basidia [(29)32–40(43) × 10.5–13.5 μm], smaller pleurocystidia [(47)52–60 × 7–10 μm] and longer cheilocystidia [(26)37–67 × 7–10 μm], slightly larger basidiospores [(14.0)15.5 ± 0.8(17.8) × (5.0)5.6 ± 0.3(6.2) μm], larger stipitipellis terminal cells [20–40 × (4)8–15(18) μm] with rare tw- spored and four-spored basidia [71]. *Xa. separans* can be easily distinguished by larger pileus [60–200(220) mm] and stipe [60–150 × 10–30 mm], longer basidia [(27)32–41(43) × 10–12 μm], smaller pleurocystidia [(52)55–70 × 11–14 μm] and larger cheilocystidia [(24)30–65(80) × 8–15(18) μm] [71].

Based on the phylogenetic analyses, the collections of *Xa. violaceipes* are clustered on one branch within *Xanthoconium* with 100% bootstrap support, and sister to *Xa. separans* with 82% statistical support, but they have a long genetic distance.

Morphologically, *Xa. violaceofuscum* has similar pileus and stipe colors to *Boletus violaceofuscus* [3,71]. However, *B. violaceofuscus* has larger basidiospores [(14.0)15.5 ± 0.8(17.8) × (5.0)5.6 ± 0.3(6.2) μm] and basidia [(29)32–40(43) × 10.5–13.5 μm], has smaller pleurocystidia [(47)52–60 × 7–10 μm], and the tubes are white or whitish at first, stipe has net–like ornamentation and at the base has white tomentum, has a few two-spored and four-spored basidia in stipitipellis, and the flesh is white.

In phylogenetic analyses, the collections of *Xa. violaceofuscum* are clustered on an independent branch within *Xanthoconium* with 100% bootstrap support.

Morphologically, *Xerocomus rutilans* is similar to *Aureoboletus zangii* [1,72] by its bright yellow hymenophore and reddish gray to grayish red pileus color. However, *Au. zangii* can be distinguished by its fox-red to English-red stipe, viscid pileus and stipe. *Au. zangii* has shorter basidia [20–30 μm], larger cheilocystidia (24–51 × 9–16 μm) and pleurocystidia [40–68(77) × 11.5–18 μm].

Phylogenetically, the samples of *Xerocomus rutilans* are clustered into an unique branch within *Xerocomus* with high statistical support (BS = 100%), and related to *X. rugosellus* [1,3] with 50% bootstrap support. However, *X. rugosellus* is characterized by the slowly bluing hymenophore and context when injured, wider basidia (12–15 μm), and t larger basidiospores [(12)14–15.5(18) × (4.5)5–5.5(7) μm] and pleuro- and cheilocystidia (45–80 × 10–14 μm).

Morphologically, *Xerocomus subsplendidus* is similar to *X. fraternus*, *X. subparvus* and *X. yunnanensis* [1]. However, *X. fraternus* can be distinguished by the changes in pileus and stipe context when injured, pileus context changing from cream to yellowish and staining bluish slowly when injured, stipe context cream on upper part, staining pale blue slowly when injured, and lower part pale red-brown near stipe base; *X. fraternus* has shorter basidiospores [(8.5)9.5–12(13) μm], and the shape of pileipellis terminal cells is subcylindrical. *X. subparvus* has cream to yellowish pileus context and a red tinge near pileipellis, staining bluish slowly or indistinctly when injured, context of the stipe is pale yellow at the upper part, staining bluish slowly when injured and pale brown to pale-red brown at the lower part; *X. subparvus* has smaller basidiospores [(8.5)9–10.5(11.5) × (3)3.5–4(4.5) μm], and the shape of pileipellis terminal cells is subcylindrical. *X. yunnanensis* is characterized by the tubes staining red-brown slowly when young and becoming bluish when mature on injury, the fresh yellow context of pileus near the pileipellis, and the white to yellowish context of stipe, bluing indistinctly when cut [1]; *X. yunnanensis* has smaller basidiospores [(9)10–11.5(13) × 4–4.5(5) μm], and the terminal cells of pileipellis are tapered or bullet-shaped.

According to our phylogeny, the sequences of *Xerocomus subsplendidus* form an independent branch within *Xerocomus* with 100% bootstrap support, and are sistered to *Xerocomus* sp. (voucher: HKAS90207, HKAS 74927 and HKAS 75076) with 58% statistical support, but they have a long genetic distance.

## Figures and Tables

**Figure 1 jof-09-00814-f001:**
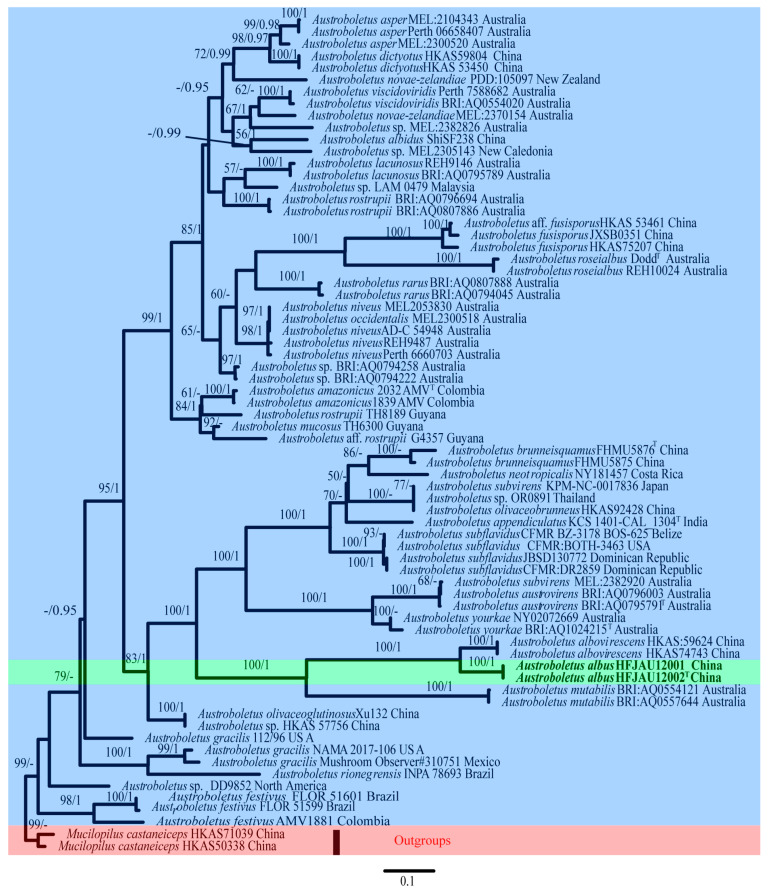
Maximum likelihood phylogenetic tree of *Austroboletus* inferred from the combined nuclear dataset (ITS + nrLSU + *TEF1-α*). Bootstrap frequencies ≥ 50% and posterior probabilities ≥ 0.95 are shown above supported branches. New sequences are shown in bold. The different colors are for decoration. The green background represents the sequence of new species discovered in this study, the red background represents the outer group, and the blue background represents other known species of this genus used in this study.

**Figure 2 jof-09-00814-f002:**
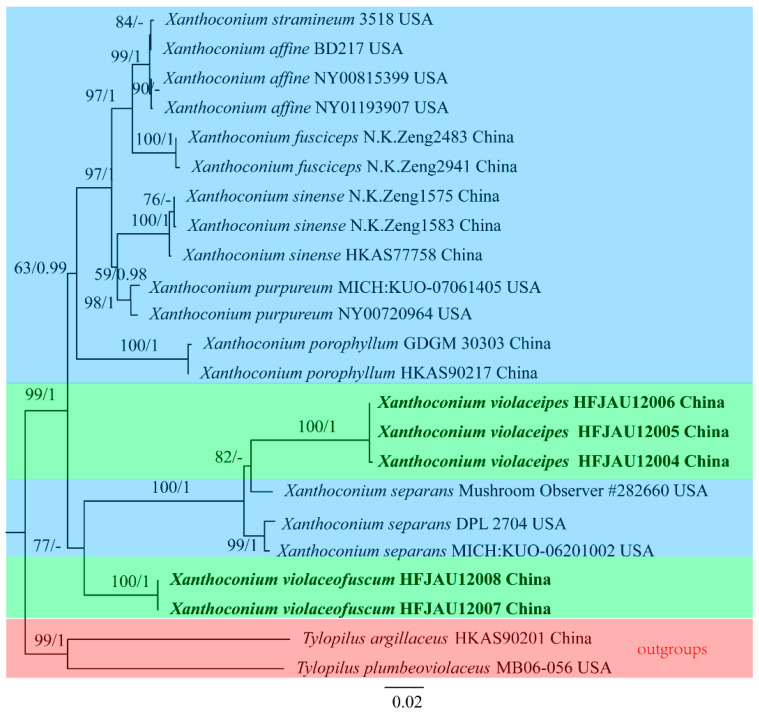
Maximum likelihood phylogenetic tree of *Xanthoconium* inferred from the combined nuclear dataset (nrLSU + *TEF1-α*). Bootstrap frequencies ≥ 50% and posterior probabilities ≥ 0.95 are shown above supported branches. New sequences are shown in bold. Different colors are used to highlight different contents. The green background represents the new species identified in this study, the blue represents other known species of the genus, and the red represents the outgroup.

**Figure 3 jof-09-00814-f003:**
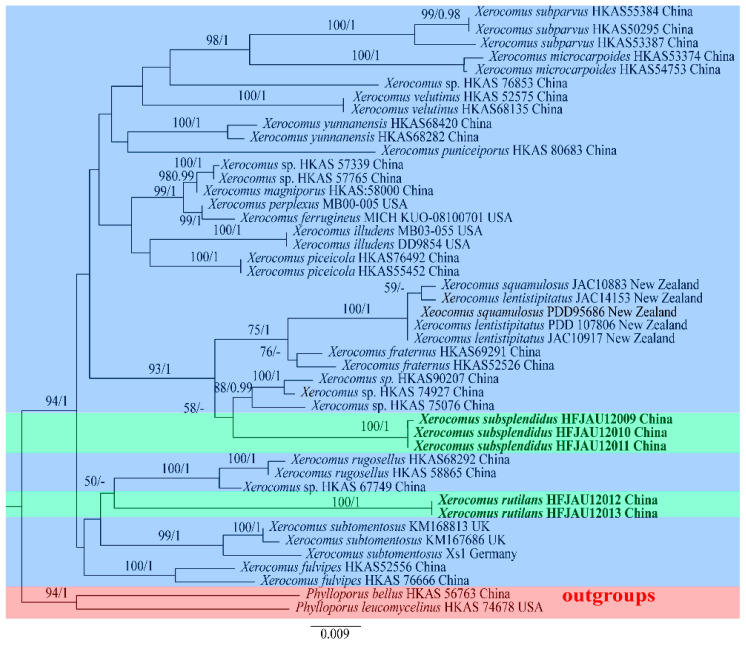
Maximum likelihood phylogenetic tree of *Xerocomus* inferred from the combined nuclear dataset (nrLSU + *TEF1-α*). Bootstrap frequencies ≥ 50% and posterior probabilities ≥ 0.95 are shown above supported branches. New sequences are shown in bold. Different colors are used to highlight different contents. The green background represents the new species identified in this study, the blue represents other known species of the genus, and the red represents the outgroup.

**Figure 4 jof-09-00814-f004:**
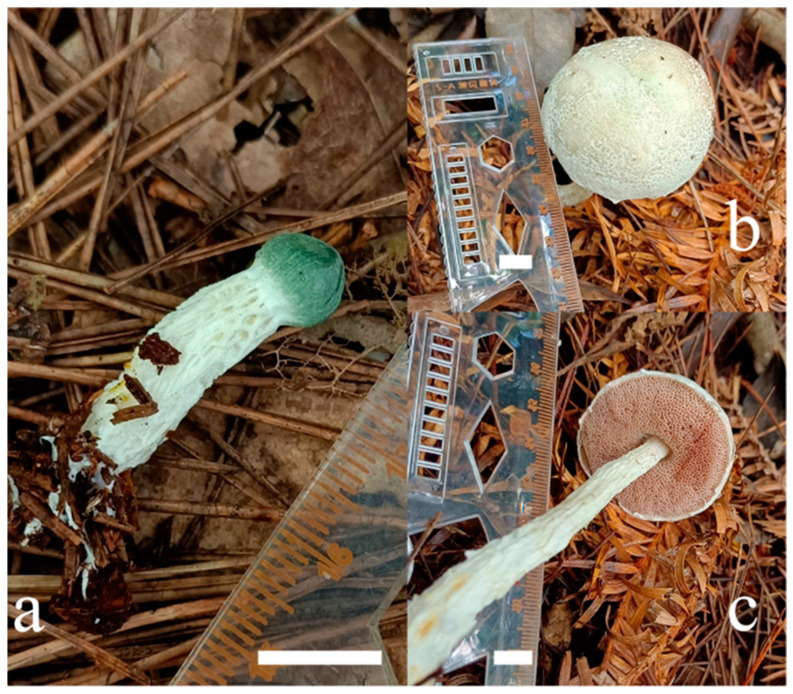
Habitat of *Austroboletus albus*. (**a**): HFJAU12001. (**b**,**c**): HFJAU12002 (Holotype). Bars = 1 cm. Photos by F. Zhou.

**Figure 5 jof-09-00814-f005:**
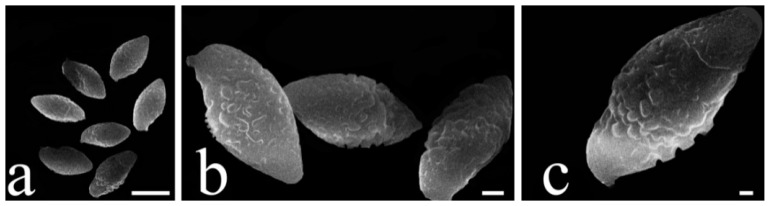
SEM of basidiospores from dried specimen of *Austroboletus albus* (HFJAU12002, Holotype). (**a**): Mag = 1.00k×, scale bars = 10 μm. (**b**): Mag = 3.00k×, scale bars = 2 μm. (**c**): Mag = 4.00k×, scale bars = 1 μm.

**Figure 6 jof-09-00814-f006:**
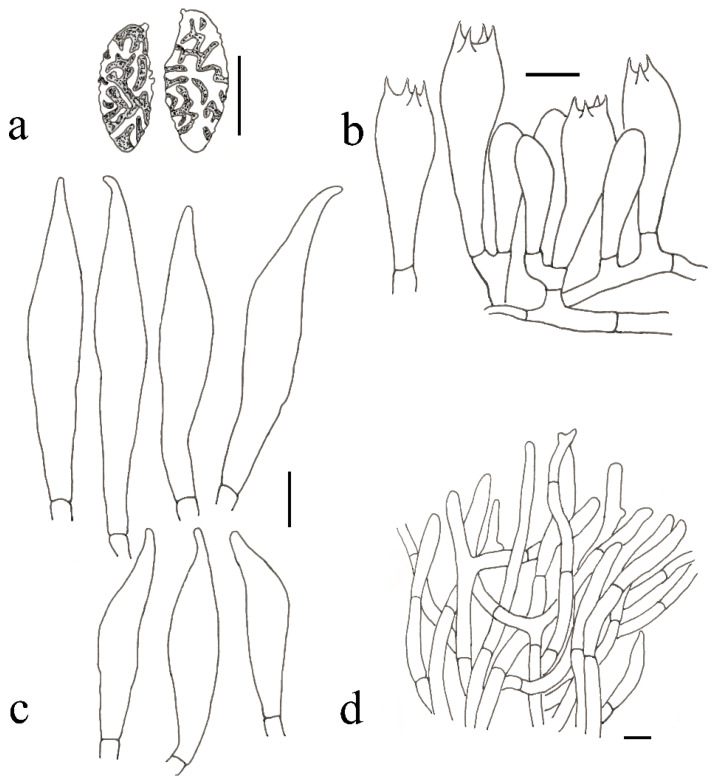
*Austroboletus albus* (HFJAU12002, Holotype). (**a**) Basidiospores. (**b**) Basidia. (**c**) Cheilocystidia. (**d**) Pileipellis. Scale bars = 10 μm. Drawings by F. Zhou.

**Figure 7 jof-09-00814-f007:**
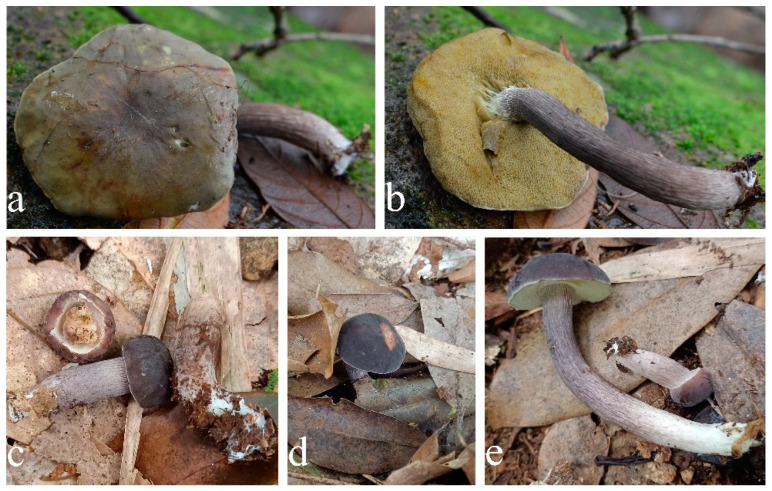
Habitat of *Xanthoconium violaceipes*. (**a**,**b**) HFJAU12004. (**c**) HFJAU12005. (**d**,**e**) HFJAU12006 (Holotype). Photos by F. Zhou.

**Figure 8 jof-09-00814-f008:**
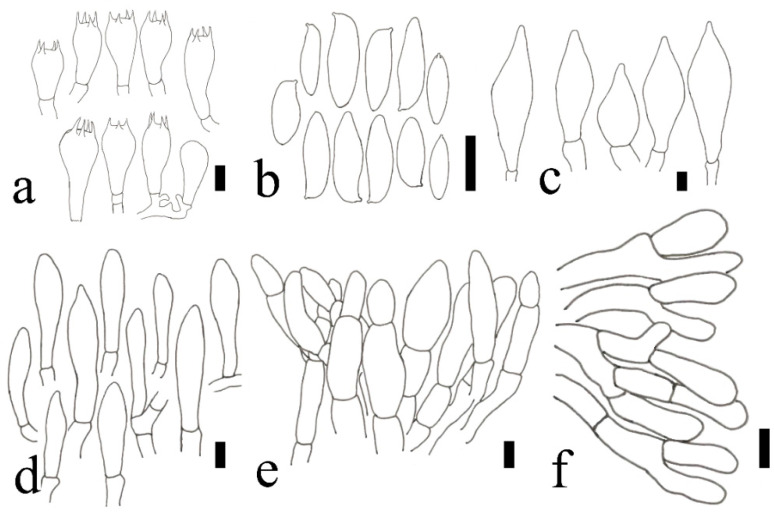
*Xanthoconium violaceipes* (HFJAU12006, Holotype). (**a**) Basidia. (**b**) Basidiospores. (**c**) Pleurocystidia. (**d**) Cheilocystidia. (**e**) Pileipellis. (**f**) Stipitipellis scale bars = 10 μm. Drawings by F. Zhou.

**Figure 9 jof-09-00814-f009:**
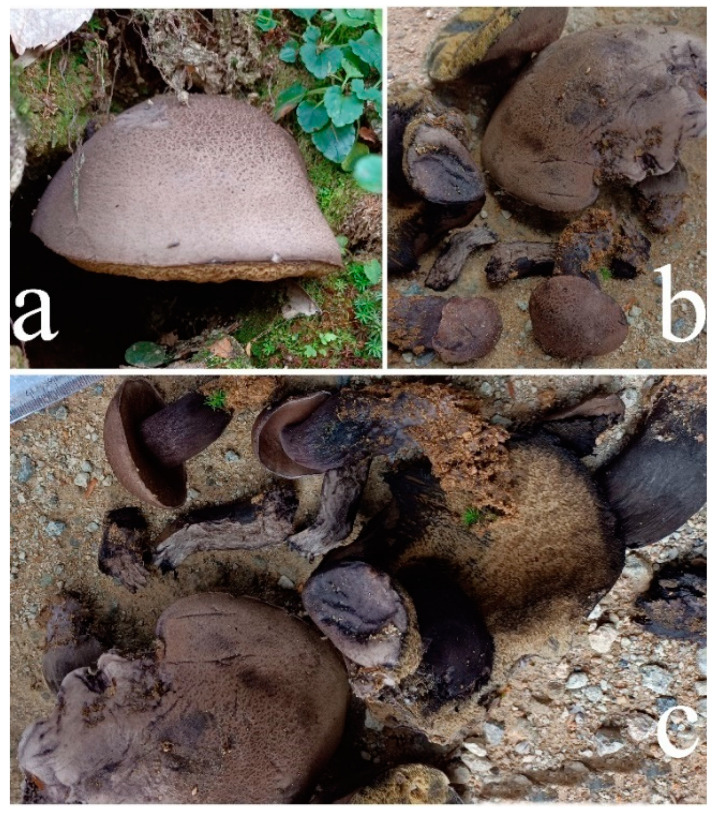
Habitat of *Xanthoconium violaceofuscum*. (**a**) HFJAU12008. (**b**,**c**) HFJAU12007 (Holotype). Photos by F. Zhou.

**Figure 10 jof-09-00814-f010:**
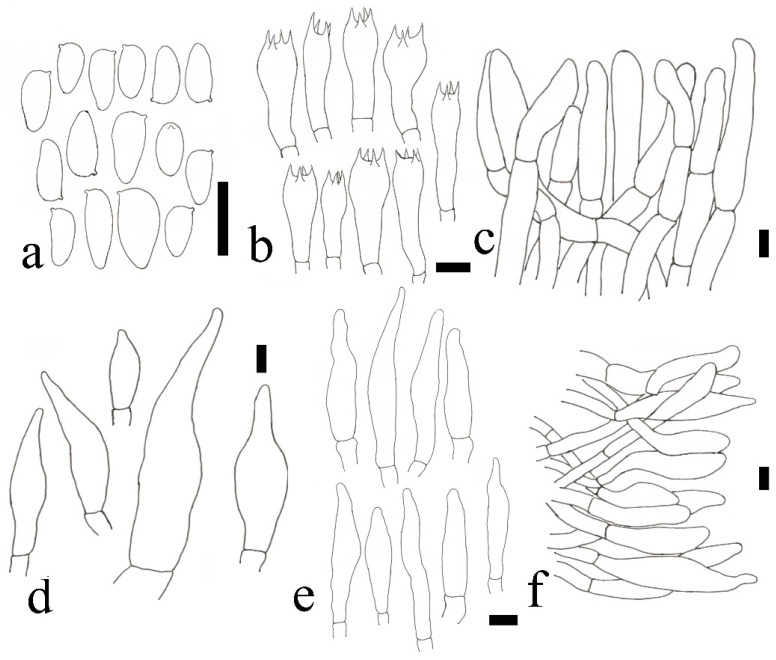
*Xanthoconium violaceofuscum* (HFJAU12007, Holotype). (**a**) Basidiospores. (**b**) Basidia. (**c**) Pileipellis. (**d**) Pleurocystidia. (**e**) Cheilocystidia. (**f**) Stipitipellis scale bars = 10 μm. Drawings by F. Zhou.

**Figure 11 jof-09-00814-f011:**
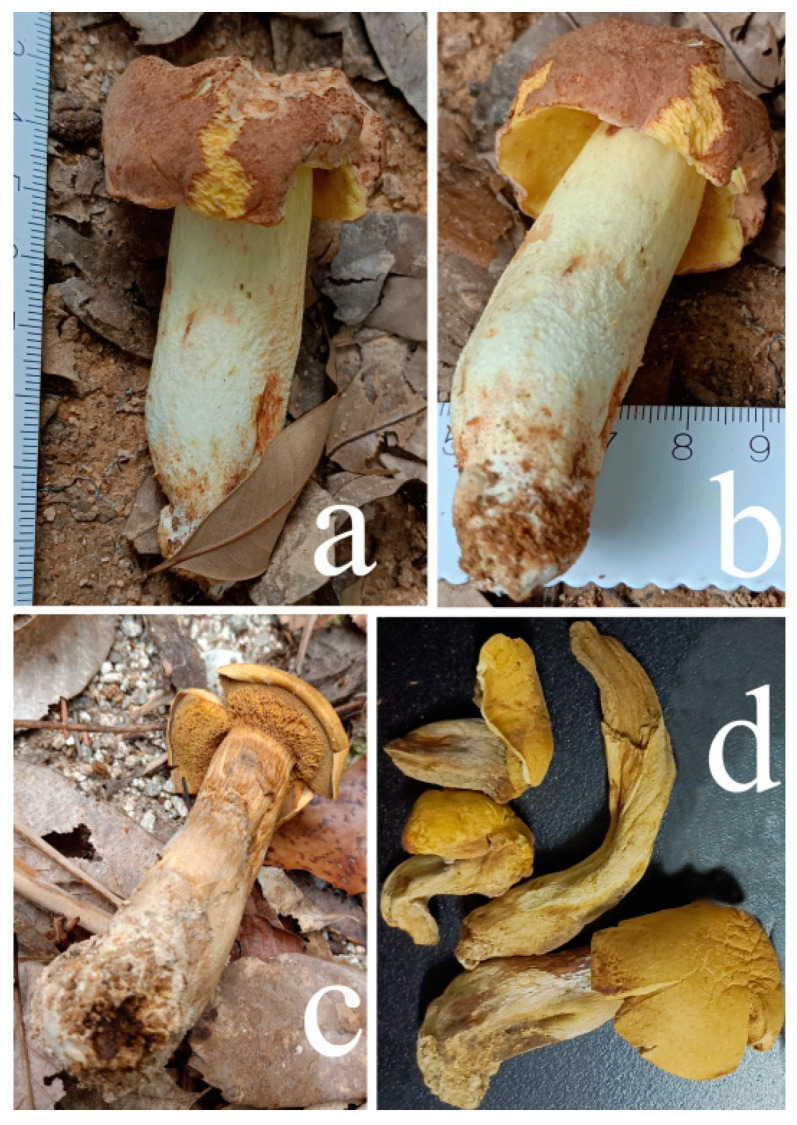
Habitat of *Xerocomus rutilans*. (**a**,**b**) HFJAU12012. (**c**,**d**) HFJAU12013(Holotype). Photos by F. Zhou.

**Figure 12 jof-09-00814-f012:**
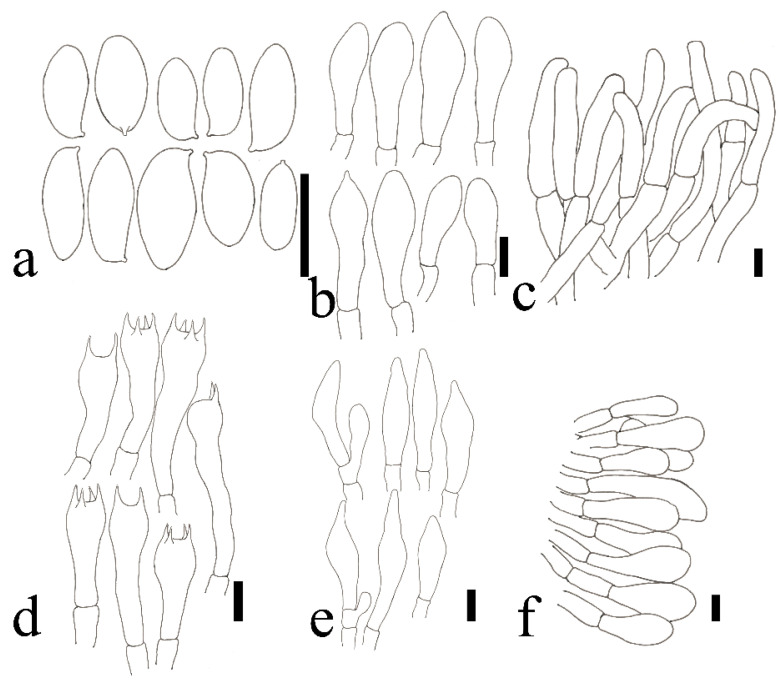
*Xerocomus rutilans*. (HFJAU12013, Holotype). (**a**) Basidiospores. (**b**) Cheilocystidia. (**c**) Pileipellis. (**d**) Basidia. (**e**) Pleurocystidia. (**f**) Stipitipellis Scale bars = 10 μm. Drawings by F. Zhou.

**Figure 13 jof-09-00814-f013:**
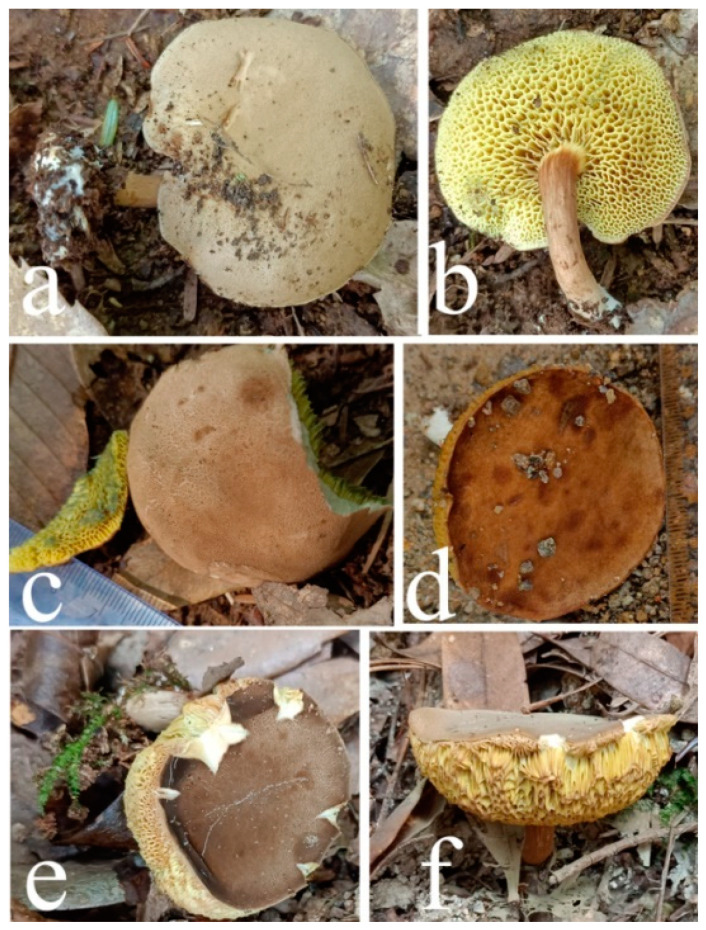
Habitat of *Xerocomus subsplendidus*. (**a**,**b**) HFJAU12009. (**c**,**d**) HFJAU12010 (Holotype). (**e**,**f**) HFJAU12011. Photos by F. Zhou.

**Figure 14 jof-09-00814-f014:**
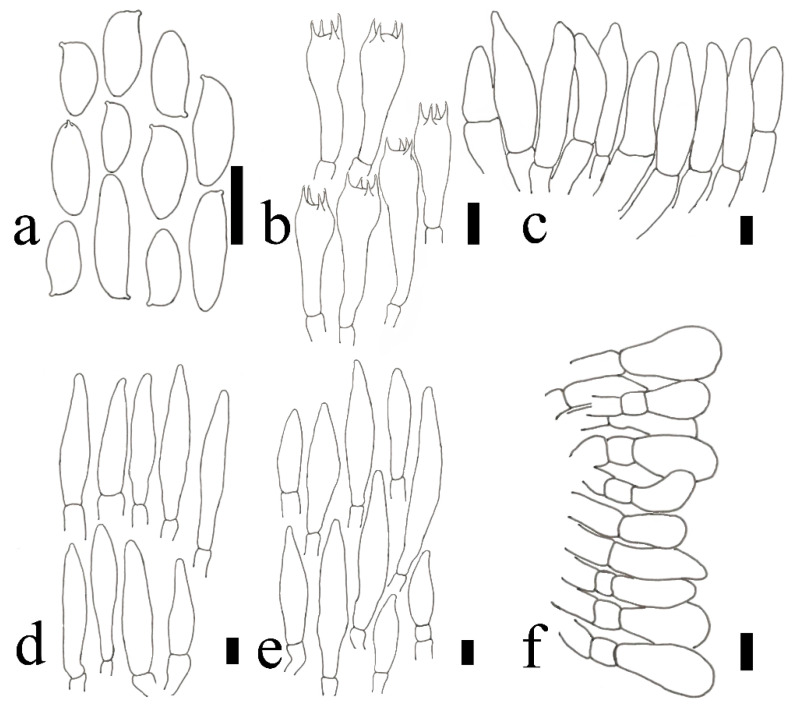
*Xerocomus subsplendidus*. (HFJAU12010, Holotype). (**a**) Basidiospores. (**b**) Basidia. (**c**) Pileipellis. (**d**) Cheilocystidia. (**e**) Pleurocystidia. (**f**) Stipitipellis Scale bars = 10 μm. Drawings by F. Zhou.

**Table 1 jof-09-00814-t001:** Names, voucher numbers, localities and corresponding GenBank accession numbers of the sequences used in phylogenetic analyses in this study. Sequences in bold were generated in this study.

Species	Voucher	Locality	GenBank Accession No.		
			ITS	LSU	*TEF1-α*
*Austroboletus*					
*Austroboletus* aff. *fusisporus*	HKAS53461	China	—	KF112486	KF112214
*Austroboletus* aff. *rostrupii*	G4357	Guyana	—	KJ786636	—
** *Austroboletus albus* **	**HFJAU12001**	**China**	**ON207028**	**ON207254**	**ON221310**
** *Austroboletus albus* **	**HFJAU12002**	**China**	**ON207029**	**ON207255**	**ON221311**
*Austroboletus albidus*	ShiSF238	China	—	MT154756	—
*Austroboletus albovirescens*	HKAS:59624	China	—	KF112485	KF112217
*Austroboletus albovirescens*	HKAS74743	China	—	KT990527	KT990730
*Austroboletus amazonicus*	2032 AMV	Colombia	KF937309	KF714510	—
*Austroboletus amazonicus*	1839 AMV	Colombia	KF937307	KF714508	—
*Austroboletus appendiculatus*	KCS 1401-CAL_1304	India	KX530028	—	—
*Austroboletus asper*	Perth 06658407	Australia	KP242216	KP242277	—
*Austroboletus asper*	MEL:2104343	Australia	KP242174	KP242260	—
*Austroboletus asper*	MEL:2300520	Australia	KP242186	KP242253	—
*Austroboletus austrovirens*	BRI:AQ0796003	Australia	KP242212	KP242228	—
*Austroboletus austrovirens*	BRI:AQ0795791	Australia	KP242211	KP242225	—
*Austroboletus brunneisquamus*	FHMU5875	China	MZ855494	MW506828	—
*Austroboletus brunneisquamus*	FHMU5876	China	MZ855495	MW506829	MW512637
*Austroboletus dictyotus*	HKAS59804	China	—	JX901138	—
*Austroboletus dictyotus*	HKAS53450	China	—	KF112487	KF112215
*Austroboletus festivus*	FLOR 51599	Brazil	KY886202	KY888001	—
*Austroboletus festivus*	FLOR 51601	Brazil	KY886203	KY888000	—
*Austroboletus festivus*	AMV1881	Colombia	KT724086	KT724095	—
*Austroboletus fusisporus*	JXSB0351	China	—	MK765810	—
*Austroboletus fusisporus*	HKAS75207	China	JX889719	JX889720	JX889718
*Austroboletus gracilis*	112/96	USA	—	DQ534624	KF030425
*Austroboletus gracilis*	NAMA 2017-106	USA	MH979242	—	—
*Austroboletus gracilis*	Mushroom Observer#310751	Mexico	MH167935	—	—
*Austroboletus lacunosus*	BRI:AQ0795789	Australia	KP242162	KP242271	—
*Austroboletus lacunosus*	REH9146	Australia	—	JX889669	JX889709
*Austroboletus mucosus*	TH6300	Guyana	—	AY612798	—
*Austroboletus mutabilis*	BRI:AQ0554121	Australia	KP242192	KP242241	—
*Austroboletus mutabilis*	BRI:AQ0557644	Australia	KP242196	KP242237	—
*Austroboletus neotropicalis*	NY181457	Costa Rica	JQ924301	JQ924334	—
*Austroboletus niveus*	AD-C 54948	Australia	KP242220	KP242280	—
*Austroboletus niveus*	Perth 6660703	Australia	KP242217	KP242279	—
*Austroboletus niveus*	MEL2053830	Australia	KC552016	KC552058	KC552099
*Austroboletus niveus*	REH9487	Australia	—	JX889668	JX889708
*Austroboletus novae-zelandiae*	PDD:105097	New Zealand	—	—	MH594051
*Austroboletus novae-zelandiae*	MEL:2370154	Australia	KP242175	KP242256	—
*Austroboletus occidentalis*	MEL2300518	Australia	KC552017	KC552059	KC552100
*Austroboletus olivaceobrunneus*	HKAS92428	China	—	—	MT110363
*Austroboletus olivaceoglutinosus*	Xu132	China	—	MT154753	MW165263
*Austroboletus rarus*	BRI:AQ0794045	Australia	KP242197	KP242236	—
*Austroboletus rarus*	BRI:AQ0807888	Australia	KP242200	—	—
*Austroboletus rionegrensis*	INPA 78693	Brazil	KY886201	—	—
*Austroboletus roseialbus*	Dodd	Australia	KY872653	KY872650	—
*Austroboletus roseialbus*	REH10024	Australia	KY872652	KY872651	—
*Austroboletus rostrupii*	BRI:AQ0807886	Australia	KP242163	KP242270	—
*Austroboletus rostrupii*	BRI:AQ0796694	Australia	KP242179	KP242258	—
*Austroboletus rostrupii*	TH8189	Guyana	JN168683	—	—
*Austroboletus* sp.	MEL:2382826	Australia	KP242213	KP242283	—
*Austroboletus* sp.	BRI:AQ0794258	Australia	KP242182	KP242255	—
*Austroboletus* sp.	BRI:AQ0794222	Australia	KP242215	KP242234	—
*Austroboletus* sp.	MEL2305143	New Caledonia	KC552018	KC552060	KC552101
*Austroboletus* sp.	HKAS57756	China	—	KF112383	KF112212
*Austroboletus* sp.	OR0891	Thailand	—	—	MH614706
*Austroboletus* sp.	LAM 0479	Malaysia	—	KY091070	—
*Austroboletus* sp.	DD9852	North America	—	AY612797	—
*Austroboletus subflavidus*	CFMR BZ-3178 BOS-625	Belize	—	MK601716	MK721070
*Austroboletus subflavidus*	JBSD130772	Dominican Republic	MT581526	MT580903	—
*Austroboletus subflavidus*	CFMR:DR2859	Dominican Republic	MT581523	MT580901	—
*Austroboletus subflavidus*	CFMR:BOTH-3463	USA	MT581521	MT580900	—
*Austroboletus subvirens*	KPM-NC-0017836	Japan	—	JN378518	JN378458
*Austroboletus subvirens*	MEL:2382920	Australia	KP012789	—	—
*Austroboletus viscidoviridis*	Perth 7588682	Australia	KP242219	KP242282	—
*Austroboletus viscidoviridis*	BRI:AQ0554020	Australia	KP242189	KP242243	—
*Austroboletus yourkae*	BRI:AQ1024215	Australia	—	MZ358814	—
*Austroboletus yourkae*	NY02072669	Australia	—	MZ358815	—
*Mucilopilus castaneiceps*	HKAS50338	China	—	KT990555	KT990755
*Mucilopilus castaneiceps*	HKAS71039	China	—	KT990547	KT990748
*Xanthoconium*					
*Xanthoconium affine*	NY00815399	USA	**/**	KT990661	KT990850
*Xanthoconium affine*	NY01193907	USA	**/**	KT990660	KT990849
*Xanthoconium affine*	BD217	USA	**/**	HQ161854	—
*Xanthoconium fusciceps*	N.K.Zeng2941	China	**/**	KY271035	—
*Xanthoconium fusciceps*	N.K.Zeng2483	China	**/**	KY271034	KY271046
*Xanthoconium porophyllum*	HKAS90217	China	**/**	KT990662	KT990851
*Xanthoconium porophyllum*	GDGM 30303	China	**/**	KC561775	—
*Xanthoconium purpureum*	NY00720964	USA	**/**	KT990663	KT990852
*Xanthoconium purpureum*	MICH:KUO-07061405	USA	**/**	MK601816	MK721170
*Xanthoconium separans*	Mushroom Observer #282660	USA	**/**	MH244206	MH347319
*Xanthoconium separans*	DPL 2704	USA	**/**	KF030329	KF030431
*Xanthoconium separans*	MICH KUO-06201002	USA	**/**	MK601723	MK721077
*Xanthoconium sinense*	N.K.Zeng1583	China	**/**	KY271032	KY271044
*Xanthoconium sinense*	HKAS77758	China	**/**	KT990665	KT990854
*Xanthoconium sinense*	N.K.Zeng1575	China	**/**	KY271031	KY271043
*Xanthoconium stramineum*	3518	USA	**/**	KF030353	KF030428
** *Xanthoconium violaceipes* **	**HFJAU12004**	**China**	**/**	**OQ146964**	**OQ162207**
** *Xanthoconium violaceipes* **	**HFJAU12005**	**China**	**/**	**OQ146965**	**OQ162208**
** *Xanthoconium violaceipes* **	**HFJAU12006**	**China**	**/**	**OQ146966**	**OQ162209**
** *Xanthoconium violaceofuscum* **	**HFJAU12007**	**China**	**/**	**OQ146967**	**OQ162210**
** *Xanthoconium violaceofuscum* **	**HFJAU12008**	**China**	**/**	**OQ146968**	**OQ162211**
*Tylopilus argillaceus*	HKAS90201	China	**/**	KT990588	KT990783
*Tylopilus plumbeoviolaceus*	MB06-056	USA	**/**	KF030350	KF030439
*Xerocomus*					
*Xerocomus ferrugineus*	MICH KUO-08100701	USA	**/**	MK601820	MK721174
*Xerocomus fraternus*	HKAS69291	China	**/**	KT990683	KT990871
*Xerocomus fraternus*	HKAS52526	China	**/**	KT990682	KT990870
*Xerocomus fulvipes*	HKAS52556	China	**/**	KT990672	KT990860
*Xerocomus fulvipes*	HKAS76666	China	**/**	KF112390	KF112292
*Xerocomus illudens*	MB03-055	USA	**/**	JQ003705	—
*Xerocomus illudens*	DD9854	USA	**/**	AY612840	—
*Xerocomus lentistipitatus*	PDD 107806	New Zealand	**/**	OP141603	—
*Xerocomus lentistipitatus*	JAC14153	New Zealand	**/**	OP141550	—
*Xerocomus lentistipitatus*	JAC10917	New Zealand	**/**	OP141509	—
*Xerocomus magniporus*	HKAS:58000	China	**/**	KF112392	KF112293
*Xerocomus microcarpoides*	HKAS54753	China	**/**	KT990680	KT990868
*Xerocomus microcarpoides*	HKAS53374	China	**/**	KT990679	KT990867
*Xerocomus perplexus*	MB00-005	USA	**/**	JQ003702	KF030438
*Xerocomus piceicola*	HKAS76492	China	**/**	KT990684	KT990872
*Xerocomus piceicola*	HKAS55452	China	**/**	KT990685	—
*Xerocomus puniceiporus*	HKAS80683	China	**/**	KU974141	KU974138
*Xerocomus rugosellus*	HKAS58865	China	**/**	KF112389	KF112294
*Xerocomus rugosellus*	HKAS68292	China	**/**	KT990686	KT990873
** *Xerocomus rutilans* **	**HFJAU12012**	**China**	**/**	**OQ146972**	**OQ162215**
** *Xerocomus rutilans* **	**HFJAU12013**	**China**	**/**	**OQ146973**	**OQ162216**
*Xerocomus squamulosus*	JAC10883	New Zealand	**/**	OP141507	—
*Xerocomus squamulosus*	PDD95686	New Zealand	**/**	JQ924327	—
*Xerocomus subparvus*	HKAS53387	China	**/**	KF112397	KF112297
*Xerocomus subparvus*	HKAS50295	China	**/**	KT990667	—
*Xerocomus subparvus*	HKAS55384	China	**/**	KT990687	KT990874
*Xerocomus subtomentosus*	KM168813	UK	**/**	KC215223	KC215249
** *Xerocomus subsplendidus* **	**HFJAU12009**	**China**	**/**	**OQ146969**	**OQ162212**
** *Xerocomus subsplendidus* **	**HFJAU12010**	**China**	**/**	**OQ146970**	**OQ162213**
** *Xerocomus subsplendidus* **	**HFJAU12011**	**China**	**/**	**OQ146971**	**OQ162214**
*Xerocomus subtomentosus*	KM167686	UK	**/**	KC215222	KC215248
*Xerocomus subtomentosus*	Xs1	Germany	**/**	AF139716	JQ327035
*Xerocomus velutinus*	HKAS68135	China	**/**	KT990673	KT990861
*Xerocomus velutinus*	HKAS52575	China	**/**	KF112393	KF112295
*Xerocomus* sp.	HKAS67749	China	**/**	KT990676	KT990864
*Xerocomus* sp.	HKAS76853	China	**/**	KF112394	KF112296
*Xerocomus* sp.	HKAS75076	China	**/**	KF112387	KF112290
*Xerocomus* sp.	HKAS57339	China	**/**	KT990674	KT990862
*Xerocomus* sp.	HKAS57765	China	**/**	KT990675	KT990863
*Xerocomus* sp.	HKAS74927	China	**/**	KF112395	KF112291
*Xerocomus* sp.	HKAS90207	China	**/**	KT990677	KT990865
*Xerocomus yunnanensis*	HKAS68420	China	**/**	KT990690	KT990877
*Xerocomus yunnanensis*	HKAS68282	China	**/**	KT990691	KT990878
*Phylloporus bellus*	HKAS56763	China	**/**	JQ967196	JQ967153
*Phylloporus leucomycelinus*	HKAS74678	USA	**/**	JQ967206	JQ967163

“**/**” means the data is not applied in this study, “—“ means the data is missing.

## Data Availability

All newly generated sequences were deposited in GenBank (https://www.ncbi.nlm.nih.gov/genbank/, Table 1 (Submitted on 30 December 2022)). All new taxa were deposited in MycoBank (https://www.mycobank.org/).

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
