# Peer review of "Phylogenetic and Morphological Evidence Reveal Five New Species of Boletes from Southern China"

_jof, 2023, doi:10.3390/jof9080814_

Round 1
Reviewer 1 Report
Dear Author (s)
This is a great piece of work on boletes of China. I have reviewed and commented minor corrections in terms of grammar and some detail question regarding the new species formation Tapinella. I think authors can improve the section accordingly. Overall its a pleasure work that you did here.

Minor corrections and grammatical errors was found. Please check and revise
Author Response
Reviewer1
We are very grateful to Reviewer for reviewing the paper so carefully. We have carefully considered the suggestions and tried our best to improve the manuscript, and all modifications have been highlighted in red font. Thank you very much!
Question 1: How many specimens were collected and deposited? Be specific.
Reply 1: Thank you so much for your suggestion, we have added it as: 12 specimens of 5 new boletus species were stored in the Herbarium of Fungi of Jiangxi Agricultural University (HFJAU).
Question 2: How many grams?
Reply 2: Thank you very much, we used approximately 0.5 g of dry specimens.
Question 3: Some incorrect words.
Reply 3: We have made modifications and see the red font in the article, thank you so much for your suggestion.
Question 4: Single specimen is not valid. Author should add more collections or discard this component.
Reply 4: Thank you for your suggestion, the relevant content of Tapinella qixianensis has been deleted.
Question 5: Change “not been observed” and “unchanging when bruised” to “absent” and “no reaction when bruised”.
Reply 5: Thank you for your suggestion, they have been changed.
Question 6: Rephrase the sentence ”Austroboletus albus is consistent with the characteristics of Austroboletus in macro and micro morphology”.
Reply 6: Thank you very much, it has been rephrase as: The macro and micro morphology of Austroboletus albus conforms to the characteristics of Austroboletus.
Question 7: Latin names should be italicized.
Reply 7: Thank you for your suggestion, those Latin name has been changed to italics.
Question 8: Submission of trees in Treebase?
Reply 8: Thank you very much, all phylogenetic trees in this study have been deposited in TreeBASE.

Reviewer 2 Report
1. Dried mushrooms may still contain contaminants that can interfere with DNA extraction and downstream applications. how did you ensure that your specimens are free from foreign materials?
2. Provide details about the sample collection process, including the number of specimens collected, would enhance the assessment of the representativeness of the newly discovered species within Jiangxi Province. You can add new title: sample collection in material and methods section.
3. Did you observe any specific habitat preferences or associations with particular tree species for the newly discovered boletus species ? add this information in taxonomy section for example
4. You have mention that gene fragments of some taxa coud not be found or sequenced and were regarded as missing data. The concatenation of sequences from two or three genes using PhyloSuite indicates an effort to increase the phylogenetic signal and resolution. However, it is important to consider the potential impact of missing data from taxa with unavailable or unsequenced gene fragments. Clarifying how missing data were handled and its potential influence on the analysis would provide a more comprehensive understanding of the study.
5. Retention of Intron Regions: The decision to retain intron regions of protein-coding genes during the analysis should be justified, as it can affect the phylogenetic inference. Provide a rationale for this choice.
6. The use of the online MAFFT version 7 website for sequence alignment is a widely accepted and reliable tool in the field. However, it would be beneficial to include specific details about the alignment settings employed, such as the chosen algorithm or any additional adjustments made.
7. The use of RAxML version 8 for Maximum Likelihood (ML) analysis and MrBayes v3.2 for Bayesian Inference (BI) are appropriate and commonly employed in phylogenetic studies. It would be helpful to specify the model of sequence evolution (GTRGAMMAI) used for ML analyses and the selected substitution models.
Author Response
Reviewer 2
We are very grateful to Reviewer for reviewing the paper so carefully. We have carefully considered the suggestions and tried our best to improve the manuscript. Thank you very much!
Question 1: Dried mushrooms may still contain contaminants that can interfere with DNA extraction and downstream applications. how did you ensure that your specimens are free from foreign materials?
Reply 1: The DNA we extracted comes from the context inside the connection between the pileus and stipe of the specimens, which is relatively clean compared to other parts and less susceptible to foreign objects.
Question 2: Provide details about the sample collection process, including the number of specimens collected, would enhance the assessment of the representativeness of the newly discovered species within Jiangxi Province. You can add new title: sample collection in material and methods section.
Reply 2: Detailed information on sample collection has been added in material and methods section.
Question 3: Did you observe any specific habitat preferences or associations with particular tree species for the newly discovered boletus species ? add this information in taxonomy section for example.
Reply 3: The newly discovered boletus species are distributed in the shade under the moist mixed broad-leaved and coniferous forests (as can be seen from the captured habitat photos). Some plants can be identified as species, but some plants can only be identified as genera or even families.
Question 4: You have mention that gene fragments of some taxa coud not be found or sequenced and were regarded as missing data. The concatenation of sequences from two or three genes using PhyloSuite indicates an effort to increase the phylogenetic signal and resolution. However, it is important to consider the potential impact of missing data from taxa with unavailable or unsequenced gene fragments. Clarifying how missing data were handled and its potential influence on the analysis would provide a more comprehensive understanding of the study.
Reply 4: Because the sequences of some species in NCBI were not sequenced, the sequences could not be found. Additionally, the ITS sequence clustering of some genera was too scattered, and the phylogenetic trees constructed from the sequences of species in the same genus could not be clustered together. Therefore, this study did not include them in the phylogenetic analysis.
Question 5: Retention of Intron Regions: The decision to retain intron regions of protein-coding genes during the analysis should be justified, as it can affect the phylogenetic inference. Provide a rationale for this choice.
Reply 5: Based on BLAST results, all available TEF1–α sequences were downloaded from NCBI(data not shown), and to evaluate the variability of the TEF1–α intron region by single-gene analyses and a later multi-gene phylogenetic analysis.
Question 6: The use of the online MAFFT version 7 website for sequence alignment is a widely accepted and reliable tool in the field. However, it would be beneficial to include specific details about the alignment settings employed, such as the chosen algorithm or any additional adjustments made.
Reply 6: The alignment of the sequence adopts the default settings without any additional adjustments.